# Optimal Assimilation Number of Phytoplankton in the Siberian Seas: Spatiotemporal Variability, Environmental Control and Estimation Using a Region-Specific Model

Andrey B. Demidov [1,*], Tatiana A. Belevich [2] 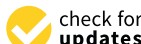 and Sergey V. Sheberstov [1]

1   Shirshov Institute of Oceanology, Russian Academy of Sciences, Moscow 117218, Russia
2   Department of Biology, Lomonosov Moscow State University, Moscow 119991, Russia
*   Correspondence: demspa@rambler.ru

**Abstract:** The maximal value of the chlorophyll-specific carbon fixation rate in the water column or the optimal assimilation number ($P^b_{opt}$) is an important parameter used to estimate water column integrated primary production (IPP) using models and satellite-derived data. The spatiotemporal variability in the $P^b_{opt}$ of the total and size-fractionated phytoplankton in the Siberian Seas (SSs) and its links with environmental factors were studied based on long-term (1993–2020) field and satellite-derived (MODIS-Aqua) observations. The average value of $P^b_{opt}$ in the SSs was equal to $1.38 \pm 0.76$ mgC (mg Chl $a$)$^{-1}$ h$^{-1}$. The monthly average values of $P^b_{opt}$ decreased during the growing season from 1.95 mgC (mg Chl $a$)$^{-1}$ h$^{-1}$ in July to 0.64 mgC (mg Chl $a$)$^{-1}$ h$^{-1}$ in October. The average value of $P^b_{opt}$ for small (<3 μm) phytoplankton 1.6-fold exceeded that for large (>3 μm) phytoplankton. The values of $P^b_{opt}$ depend mainly on incident photosynthetically available radiation (PAR). Based on the relationship between $P^b_{opt}$ and PAR, the empirical region-specific algorithm ($E_{0reg}$) was developed. The $E_{0reg}$ algorithm performed better than commonly used temperature-based models. The application of $E_{0reg}$ for the calculation of $P^b_{opt}$ will make it possible to more precisely estimate IPP in the SSs.

**Keywords:** optimal assimilation number; primary production; chlorophyll $a$; size-fractionated phytoplankton; remote sensing; Siberian Seas

## 1. Introduction

The primary production (PP) of oceanic phytoplankton amounts to approximately half of the net autotrophic production of the Earth [1]. PP is an important factor in $CO_2$ exchange between the atmosphere and ocean, which is one of the factors that determine global climate change [2–7]. Therefore, the estimation of the long-term variability in PP under climate trends is one of the main tasks of the biogeochemistry of the World Ocean [8]. Currently, this estimation is carried out using production and biogeochemical models with satellite-derived data as input variables.

Parametrisation is one of the main problems in PP modelling and one of the main factors determining the model performance. Chlorophyll $a$ (Chl $a$)-normalised PP (chlorophyll-specific carbon fixation rate or assimilation number; $P^b$ = PP/Chl $a$) is the most important parameter characterising the photoadaptive processes of phytoplankton, used to develop primary production algorithms and estimate the spatiotemporal variations in PP. There are two approaches to $P^b$ determination. One of them is P-I experiments, which establish the relationship between the rate of carbon assimilation and the intensity of artificial light during short (≤4 h) exposions described using the models fitted to the photosynthesis vs. irradiance curves [9,10]. Such experiments are carried out in photosynthetrons where irradiance saturating of photosynthesis is achieved. The value of the chlorophyll-specific carbon fixation rate at saturating light intensity is defined as the maximal assimilation number ($P^b_{max}$). With a different approach, $P^b$ is estimated during measurements of integral

PP in the water column (IPP) under natural illumination [11,12]. Therein, the maximal $P^b$ is determined at the depth with optimal irradiance for photosynthesis and defined as the optimal assimilation number ($P^b_{opt}$). It should be noted that under natural conditions, the maximal values of carbon assimilation can be achieved at a light intensity less than that which saturates photosynthesis. Therefore, the parameters of $P^b_{max}$ and $P^b_{opt}$ are not equivalent [13].

The present paper studies the spatiotemporal variations in $P^b_{opt}$ and its links with environmental variables. This parameter is widely used in so-called chlorophyll-based PP models [13,14]. The comparison of the predictive skill of these algorithms with other models was repeatedly carried out previously [15–24].

The development and validation of PP models for the assessment of IPP in the Arctic Ocean is a complicated problem. The Arctic Ocean is an under-sampled region in terms of in situ PP measurements, which is one of the components of this problem. Therefore, prominent evaluations of the Arctic Ocean IPP were performed using the models, which were originally developed for other parts of the World Ocean [21]. This approach decreases the accuracy of IPP estimation in the Arctic Ocean. Meanwhile, it is known that using regional-specific algorithms increases the efficiency of IPP assessment [18,21,25–27].

The Siberian Seas (SSs), which include the Kara, Laptev, and East Siberian Seas, are the least studied among all areas of the Arctic Ocean in terms of PP processes [28–30]. Thus, little is known about the values of $P^b_{opt}$ in the SSs, its relationships with environmental factors, and the magnitude of $P^b_{opt}$ of size-fractionated phytoplankton [31–33]. Meanwhile, it is known that size composition is an important abiotic factor affecting the $P^b_{opt}$ value of phytoplankton [34].

Determination of the range in variability and the average value of $P^b_{opt}$ in the SSs is critically important for the investigation of PP features in this region. These features are linked with the particularity of the phytoplankton biotope that functions on the broad continental shelf under the influence of intense river runoff [35–39]. Freshwater discharge into the Siberian shelf leads to low salinity in the subsurface layer, sharp stratification [40–43], and high particulate (POM) and coloured dissolved (CDOM) organic matter, as well as the concentration of terrigenous mineral suspension [35,44–46]. Consequently, the Kara Sea waters are characterised by high turbidity, low transparency (average Secchi disk depth of 8 m), and a small photosynthetic layer (22 m on average) [47]. Therefore, it seems relevant to develop the region-specific algorithm of $P^b_{opt}$ for the SSs as one of the main parameters of PP models.

There are two approaches to the application of $P^b_{opt}$ in PP models. According to one of them, $P^b_{opt}$ is used as the average value for a particular biogeochemical province [48,49]. The second approach assumes the calculation of $P^b_{opt}$ by its relationship with a value of an environmental factor that is determined with remote sensing from space. The second one is recommended to be used in areas of the World Ocean with a high spatiotemporal variability in biogeochemical parameters [50]. The intense river runoff is the reason for sharp spatial gradients of hydrophysical, hydrochemical, and biological parameters in the SSs. Therefore, the application of the second approach to $P^b_{opt}$ modelling can improve the model performance. To implement this method, it is necessary to establish the relationships between $P^b_{opt}$ and the environmental factors: photosynthetically available radiation (PAR), nutrient concentration, water temperature, salinity, and Chl *a* concentration.

Thus, for region-specific modelling, it is relevant to develop an empirical algorithm describing the relationships between $P^b_{opt}$ and environmental factors. The main abiotic variable that determines $P^b_{opt}$ values and that is easily assessed using remote sensing is sea surface temperature ($T_0$). Meanwhile, it is known that other environmental factors limiting the rate of photosynthesis such as PAR and nutrients constrain IPP and $P^b_{opt}$ at high latitudes [51–55]. Here, it is postulated that the PAR-based $P^b_{opt}$ algorithm is more effective in the SSs than the T-based models. The development of a sufficiently effective region-specific model of $P^b_{opt}$ will make it possible in the future to obtain new estimates of the annual values of IPP in the SSs using satellite-derived data.

Thus, the aims of the present article were: (1) to establish the ranges in variability and the average values of $P^b_{opt}$ in the SSs; (2) to evaluate the influence of the environmental factors on $P^b_{opt}$; and (3) to develop an empirical region-specific model and apply it to describe the spatial distribution in $P^b_{opt}$ in the SSs using satellite-derived data.

## 2. Materials and Methods

### 2.1. Data Sources and Sampling

The field data were obtained in boreal summer (July, August) and autumn (September, October) during 11 cruises in the Siberian Seas (SSs) in 1993, 2007, 2011, and 2013–2020 (Table 1). The sampling sites where the measurements of primary production (PP), chlorophyll *a* concentration (Chl *a*) of total and size-fractionated phytoplankton, and environmental parameters were performed are shown in Figure 1. At these sites, the calculations of the optimal assimilation number ($P^b_{opt}$) were performed. The values of the measured variables are shown in the Supplementary Materials (Table S1).

**Table 1.** Sources for primary production and chlorophyll *a* measurements included in the dataset for analysis of the variability in the optimal assimilation number, its links with environmental factors, and model development and verification.

| Cruise | Months | Years | Location | Number of Stations | Publications |
|---|---|---|---|---|---|
| 49th Dmitry Mendeleev | August–September | 1993 | Kara Sea | 29 | [56] |
| 54th Akademik Mstislav Keldysh | September | 2007 | Kara Sea | 16 | [57] |
| 59th Akademik Mstislav Keldysh | September–October | 2011 | Kara Sea | 36 | [58] |
| 125th Professor Shtokman | September | 2013 | Kara Sea | 29 | [59] |
| 128th Professor Shtokman | August–September | 2014 | Kara Sea | 48 | Unpublished data |
| 63d Akademik Mstislav Keldysh | August–October | 2015 | Kara and Laptev Seas | 56 | [60] |
| 66th Akademik Mstislav Keldysh | July–August | 2016 | Kara Sea | 55 | [61] |
| 69th Akademik Mstislav Keldysh | August–September | 2017 | Kara, Laptev, and East Siberian Seas | 53 | [60,62] |
| 72nd Akademik Mstislav Keldysh | August–September | 2018 | Kara and Laptev Seas | 37 | [60] |
| 76th Akademik Mstislav Keldysh | July–August | 2019 | Kara Sea | 32 | Unpublished data |
| 81st Akademik Mstislav Keldysh | August–September | 2020 | Kara Sea | 28 | Unpublished data |

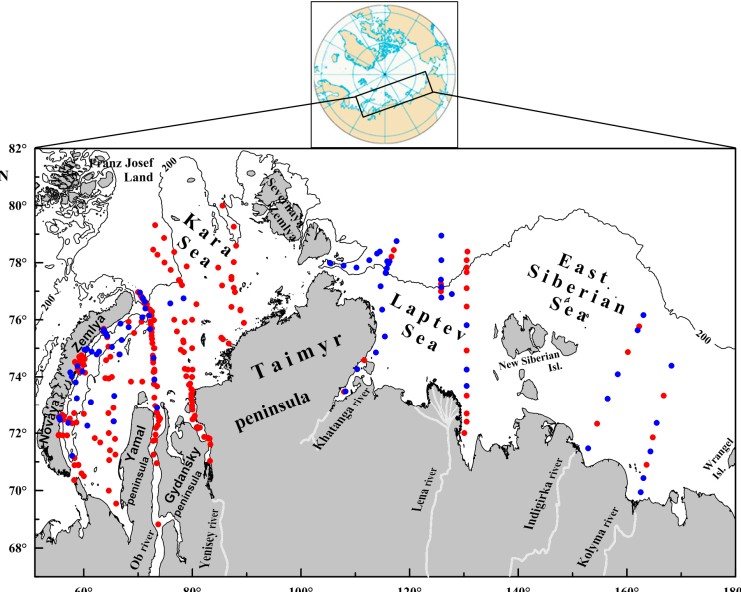

**Figure 1.** Locations of the sampling sites in the Siberian Seas where the calculations of the optimal assimilation number ($P^b_{opt}$) were performed. The red circles indicate the sites where the calculations of $P^b_{opt}$ of the total phytoplankton were carried out. The blue circles indicate the sites where the calculations of $P^b_{opt}$ of size-fractionated phytoplankton were performed.

The sampling depths were defined after a preliminary sounding of temperature, conductivity, and chlorophyll fluorescence using a CTD probe SBE-19 and SBE-32 (Seabird Electronics Inc., Bellevue, WA, USA). Niskin bottles were deployed at the stations to obtain water samples from discrete depths within the upper 100 m layer. Trace metal cleaning procedures (e.g., Teflon-coated covers and springs for the Niskin bottles) [63] were used during all the cruises.

### 2.2. The Field Data

The methods for determining PP and Chl *a* are described in detail in previous studies [47,57]. PP was estimated on board using a radiocarbon technique [64] according to simulated in situ approach. Acid-cleaned 160 mL bottles with water samples after the addition of sodium bicarbonate ($NaH_{14}CO_3$, 0.05 µCi per 1 mL of sample) were placed under neutral lighting filters and exposed for half of a light day in a deck incubator with the seawater temperature maintained at the in situ conditions. The transparency of neutral lighting filters was chosen based on the light exposure conditions at the sampling depths after the sounding of underwater photosynthetically available radiation (PAR). After exposure, the samples were filtered onto a 0.45 µm nitrocellulose membrane "Vladipore" (Vladipore, Vladimir, Russia). After filtration, the samples were treated with 0.1 N HCl and filtered seawater, dried overnight, and placed in a scintillation vial with 10 mL of the scintillation cocktail "Optiphase HiSafe III" (PerkinElmer, Waltham, MA, USA). The radioactivity in the samples was determined after 24 h using a liquid scintillation counter "Triathler" (Hidex, Turku, Finland).

The Chl *a* concentration was determined using a spectrophotometric method [65,66] or fluorometrically [67,68]. Previous comparisons have shown good agreement between various methods of Chl *a* determination [69,70]. The PP and Chl *a* data that were obtained with these methods were used for $P^b_{opt}$ calculations. The $P^b_{opt}$ value was determined as the maximal value of the PP to Chl *a* ratio in the water column.

The intensity of the incident surface irradiance was measured with an LI-190SA (LI-COR) sensor [58–62]. The daily PAR was obtained from integration in the LI-1400 module for five-minute intervals (mol quanta $m^{-2}$) and saved in the internal memory. Underwater irradiance was measured in the following mode. The LI-192SA underwater light sensor, mounted vertically on a cable and in the sounding mode, was moved down to a depth of ∼60–80 m and, at shallow stations, down to the bottom.

Concentrations of silicates ($Si(OH)_4$), phosphates ($PO_4$), nitrites ($NO_2$), nitrates ($NO_3$), and ammonium ($NH_4$) were measured using the methods described previously [71,72]. Colourimetric determinations were performed with HACH Lange DR 2800 and LEKI SS2107UV spectrophotometers. Determination of the total alkalinity (Alk) was carried out using the direct titration technique. Calculations of the dissolved $CO_2$ and concentrations of various forms of dissolved inorganic carbon were performed with the pH-Alk method using thermodynamic equations for the carbon balance with constants for carbonic acid dissociation [72,73].

The values of environmental variables at the depth with $P^b_{opt}$ were used for statistical analysis. It should be noted that the $P^b_{opt}$ values were predominantly observed within the upper 0–2 m layer (Table S1).

To determine Chl *a* and PP of large phytoplankton (>3 µm), samples were successively filtered through a Nucleopore filter with a 3 µm pore size (Reatrack, Obninsk, Russia). Small (<3 µm) Chl *a* and PP values were obtained by subtracting the large phytoplankton from the total Chl *a* and PP values [33].

The spatial variability in PP characteristics in the Siberian Seas (SSs) depends mainly on the distribution of river runoff [58–62]. Therefore, it can be assumed that the values of $P^b_{opt}$ in the areas under the influence of riverine waters and the areas without such impact can be different. Sea surface salinity ($S_0$) is the indicator of these types of waters ($S_0 < 25$ and >25, respectively). Thus, the average values of $P^b_{opt}$ for the regions with $S_0 < 25$ and >25 were calculated separately. According to [74], the annual average isohaline 25 separates

brackish waters and waters with salinity close to oceanic. Water salinity was measured using the Practical Salinity Scale.

### 2.3. Statistical Analysis

Before calculations, data were log-transformed to achieve normal distribution and for use in the parametrical statistic methods. Then, data were checked for normality using the Kolmogorov–Smirnov test (Figure S1).

The relationships between parameters were tested using linear regression and principal component analysis (PCA). Correspondences between log-transformed variables were estimated using Pearson's coefficient of correlation (R). A difference between sample means was assessed using Student's *t*-test. The null hypothesis was rejected at $p < 0.01$. Statistical calculations were performed using the Statistica 6.0 software (StatSoft Inc., Tulsa, OK, USA).

### 2.4. Development and Verification of $P^b_{opt}$ Models

The entire dataset was randomly divided into two parts. Two-thirds and one-third were used for model development and validation, respectively. The relationships between the measured and modelled $P^b_{opt}$ estimates were tested using linear regression. The variance in the dependent values was defined by the coefficient of determination ($R^2$). The slope and intercept of the linear regression determined the fitted line according to a 1:1 agreement.

The root-mean-square difference (RMSD) was used to assess the model performance. The RMSD revealed differences between the log-transformed measured and modelled values and comprised both bias (systematic error) and variability ($\sigma$—random error) [75,76]. The log-normalised RMSD was used to assess the overall model performance in Primary Productivity Algorithm Round Robins (PPARR) studies [15,17–19,21]. The models with lower RMSD have higher skill and vice versa. An RMSD value close to 0.3 indicates model over- or underestimation by a factor of 2. In addition, the mean bias (B) of each model was calculated to assess over- or underestimated $P^b_{opt}$.

### 2.5. Satellite-Derived Data of Photosynthetically Available Radiation (PAR)

Moderate Resolution Imaging Spectroradiometer (MODIS-Aqua) Level 2 data on PAR with $9 \times 9$ km resolution were obtained from NASA's Goddard Space Flight Centre (NASA GSFC) (www.oceancolor.gsfc.nasa.gov/ (accessed on 22 August 2022)). Data on PAR were used as a standard product of the MODIS-Aqua scanner [77]. The time period of the satellite data coincided with in situ observations (2007–2020). In situ and satellite data are considered to be matched up on the same day. All the satellite-derived data products were calculated as average values over acceptable nine pixels around a given point (in situ and satellite match-up sites, $N = 373$ for $P^b_{opt}$ and $N = 322$ for PAR). A pixel was considered acceptable if it was without flags of cloudiness or land. The lists of matched-up in situ and satellite data are represented in Tables S2 and S3.

## 3. Results

### 3.1. Values and Spatiotemporal Variability in Optimal Assimilation Number ($P^b_{opt}$) in the Siberian Seas (SSs) Using Field Observations

The total values of $P^b_{opt}$ in the SSs changed from 0.11 to 4.67 mgC (mg Chl *a*)$^{-1}$ h$^{-1}$. The average values varied from $1.27 \pm 0.58$ mgC (mg Chl *a*)$^{-1}$ h$^{-1}$ in the Laptev Sea to $1.77 \pm 0.70$ mgC (mg Chl *a*)$^{-1}$ h$^{-1}$ in the East Siberian Sea. The average value of $P^b_{opt}$ in the SSs was $1.38 \pm 0.76$ mgC (mg Chl *a*)$^{-1}$ h$^{-1}$ (Table 2).

The range of $P^b_{opt}$ variability in large phytoplankton (>3 μm) ($P^b_{opt L}$) was 0.23–3.54 mgC (mg Chl *a*)$^{-1}$ h$^{-1}$. The average values of $P^b_{opt L}$ varied insignificantly from $1.02 \pm 0.55$ mgC (mg Chl *a*)$^{-1}$ h$^{-1}$ in the Laptev Sea to $1.21 \pm 0.68$ mgC (mg Chl *a*)$^{-1}$ h$^{-1}$ in the East Siberian Sea (Table 2). The total values of $P^b_{opt}$ for small phytoplankton (<3 μm) ($P^b_{opt S}$) were more variable than $P^b_{opt L}$ and changed from 0.03 to 6.38 mgC (mg Chl *a*)$^{-1}$ h$^{-1}$. The



average value of $P^b_{opt\,S}$ was the highest in the Kara Sea (1.78 ± 1.29 mgC (mg Chl $a$)$^{-1}$ h$^{-1}$), and it was the lowest in the Laptev Sea (1.30 ± 0.76 mgC (mg Chl $a$)$^{-1}$ h$^{-1}$).The average value of $P^b_{opt\,S}$ in the SSs was 1.6-fold higher than $P^b_{opt\,L}$ (Table 2). The difference between the average values of $P^b_{opt\,S}$ and $P^b_{opt\,L}$ was statistically significant (Student's $t$-test, $p < 0.01$).

**Table 2.** The variability in the optimal assimilation number (mgC (mgChl $a$))$^{-1}$ h$^{-1}$ of different phytoplankton size groups in the Siberian Seas. $M$—mean; σ—standard deviation; $N$—number of data.

| Region | Statistics | >3 μm | <3 μm | Total |
|---|---|---|---|---|
| Kara Sea | $M \pm \sigma$ | 1.15 ± 0.59 | 1.78 ± 1.29 | 1.37 ± 0.79 |
| | $N$ | 60 | 59 | 333 |
| Laptev Sea | $M \pm \sigma$ | 1.02 ± 0.55 | 1.30 ± 0.76 | 1.27 ± 0.58 |
| | $N$ | 33 | 33 | 59 |
| East Siberian Sea | $M \pm \sigma$ | 1.21 ± 0.68 | 1.33 ± 0.97 | 1.77 ± 0.70 |
| | $N$ | 12 | 12 | 19 |
| Siberian Seas | $M \pm \sigma$ | 1.05 ± 0.58 | 1.65 ± 1.24 | 1.38 ± 0.76 |
| | $N$ | 105 | 104 | 411 |

The difference between the average values of $P^b_{opt}$, $P^b_{opt\,L}$, and $P^b_{opt\,S}$ in the river runoff regions with surface salinity ($S_0$) < 25 and in the areas out of the river's influence ($S_0 > 25$) was statistically insignificant. In addition, the seasonal average values of $P^b_{opt}$, $P^b_{opt\,L}$, and $P^b_{opt\,S}$ in these regions differentiated slightly (Table 3).

**Table 3.** The optimal assimilation number (mgC (mgChl $a$))$^{-1}$ h$^{-1}$ of different phytoplankton size groups within the different salinity ranges and seasons. $S_0$—sea surface salinity; $M$—mean; σ—standard deviation; $N$—number of data.

| Range of $S_0$ | Season | Statistics | Phytoplankton Size Fractions | | | | | |
|---|---|---|---|---|---|---|---|---|
| | | | >3 μm | | <3 μm | | Total | |
| <25 | Summer (July, August) | $M \pm \sigma$ | 1.26 ± 0.43 | | 2.60 ± 1.96 | | 1.72 ± 0.48 | |
| | | $N$ | 3 | 1.10 ± 0.52 | 3 | 1.73 ± 1.40 | 41 | 1.35 ± 0.64 |
| | Autumn (September, October) | $M \pm \sigma$ | 1.07 ± 0.54 | | 1.60 ± 1.32 | | 1.22 ± 0.57 | |
| | | $N$ | 21 | | 21 | | 117 | |
| >25 | Summer (July, August) | $M \pm \sigma$ | 1.13 ± 0.66 | | 1.99 ± 1.42 | | 1.81 ± 0.88 | |
| | | $N$ | 35 | 1.04 ± 0.60 | 35 | 1.63 ± 1.19 | 110 | 1.40 ± 0.84 |
| | Autumn (September, October) | $M \pm \sigma$ | 0.97 ± 0.54 | | 1.35 ± 0.90 | | 1.08 ± 0.64 | |
| | | $N$ | 46 | | 45 | | 143 | |

In the summer, the average values of $P^b_{opt}$ exceeded those in the autumn over the regions with $S_0 < 25$ and $S_0 > 25$ by factors of 1.4 and 1.7, respectively. These differences were statistically significant (Student's $t$-test, $p < 0.01$). In the summer, the average values of $P^b_{opt}$ for different size fractions of phytoplankton were higher than in the autumn both in the brackish and in the oceanic waters (Table 3). It should be noted that the difference was statistically significant only for $P^b_{opt\,S}$ at $S_0 > 25$. The monthly average values of $P^b_{opt}$ decreased during the growing season from 1.95 mgC (mg Chl $a$)$^{-1}$ h$^{-1}$ in July to 0.64 mgC (mg Chl $a$)$^{-1}$ h$^{-1}$ in October following the monthly average values of subsurface photosynthetically available radiation (PAR) ($E_0$) and sea surface temperature ($T_0$) (Figure 2).

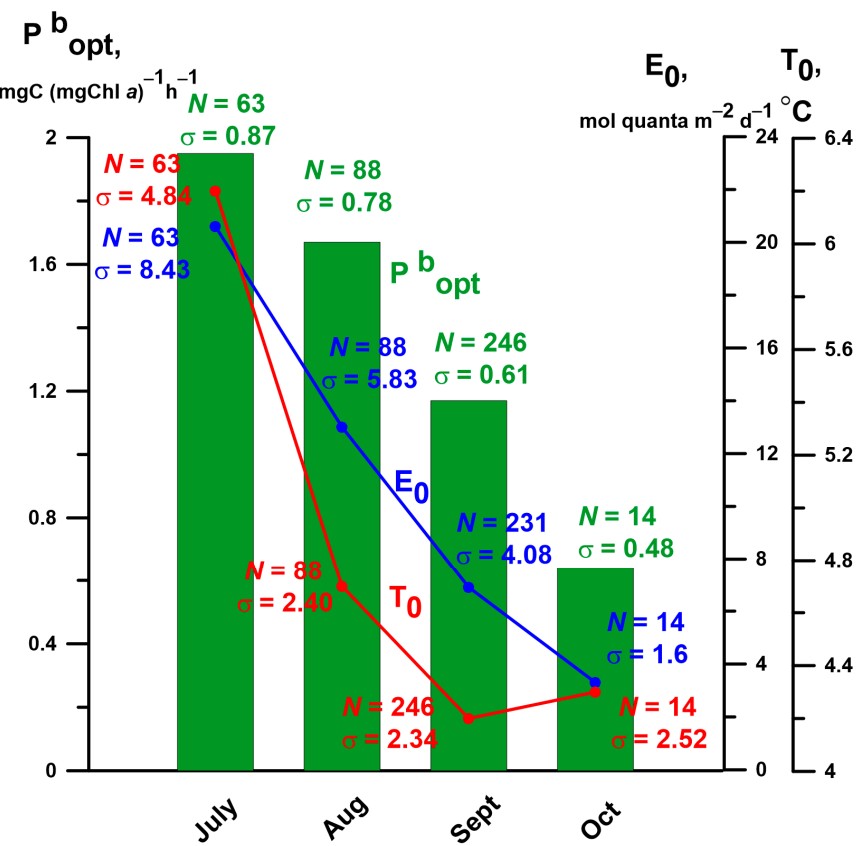

**Figure 2.** Seasonal variation in the optimal assimilation number and associated environmental factors in the Siberian Seas. Bars are the average values of the optimal assimilation number ($P^b_{opt}$). The blue line is subsurface PAR ($E_0$). The red line is sea surface temperature ($T_0$). The vertical line segments indicate standard deviation.

### 3.2. The Relationships between $P^b_{opt}$ and Environmental Factors

The relationships between $P^b_{opt}$ and environmental factors in different seasons are shown in Figure 3. The results of the correlation analysis are represented in Table 4. The statistically significant positive link with a high coefficient of correlation (R = 0.61) was established between $P^b_{opt}$ and $E_0$. There were weak links between $P^b_{opt}$ and the other environmental variables (Table 4).

**Table 4.** The correlation matrix between the log-transformed optimal assimilation numbers for different phytoplankton size groups and environmental variables. $P^b_{opt}$—optimal assimilation number of the total phytoplankton; $P^b_{opt\,L}$—optimal assimilation number of large phytoplankton (>3 μm); $P^b_{opt\,S}$—optimal assimilation number of small phytoplankton (<3 μm); R—coefficient of correlation; *p* value—statistical significance of R; *N*—the number of data. $T_0$—sea surface temperature; $S_0$—sea surface salinity; $PO_4$, $Si(OH)_4$, and DIN–surface concentrations of phosphates, dissolved silicon, and dissolved inorganic nitrogen, respectively; $Chl_0$—chlorophyll *a* concentration of the surface total phytoplankton; $Chl_S$—chlorophyll *a* concentration of surface small (<3 μm) phytoplankton; $E_0$—subsurface photosynthetically available radiation. The asterisks indicate significant correlations (*p* < 0.05).

| Parameter | Statistics | $T_0$ | $S_0$ | $Chl_0$ | $Chl_S/Chl_0$ | $E_0$ | $PO_4$ | $Si(OH)_4$ | DIN |
|---|---|---|---|---|---|---|---|---|---|
| | R | 0.08 | 0.08 | −0.21 * | 0.23 * | 0.61 * | −0.02 | 0.10 | −0.20 * |
| $P^b_{opt}$ | *N* | 411 | 408 | 411 | 104 | 395 | 404 | 410 | 387 |
| | *p* | 0.109 | 0.092 | $<10^{-3}$ | 0.020 | $<10^{-2}$ | 0.723 | 0.12 | $<10^{-3}$ |

**Table 4.** *Cont.*

| Parameter | Statistics | $T_0$ | $S_0$ | $Chl_0$ | $Chl_S/Chl_0$ | $E_0$ | $PO_4$ | $Si(OH)_4$ | DIN |
|---|---|---|---|---|---|---|---|---|---|
| $P^b_{opt\ L}$ | R | 0.20 * | 0.07 | 0.01 | 0.26* | 0.08 | 0.14 | 0.16 | 0.13 |
| | N | 105 | 105 | 105 | 104 | 105 | 104 | 104 | 98 |
| | p | 0.04 | 0.468 | 0.947 | 0.008 | 0.394 | 0.154 | 0.106 | 0.201 |
| $P^b_{opt\ S}$ | R | 0.07 | −0.04 | 0.12 | −0.09 | 0.24 * | −0.11 | −0.05 | −0.05 |
| | N | 104 | 104 | 104 | 103 | 104 | 103 | 103 | 97 |
| | p | 0.457 | 0.652 | 0.208 | 0.379 | 0.014 | 0.291 | 0.594 | 0.596 |

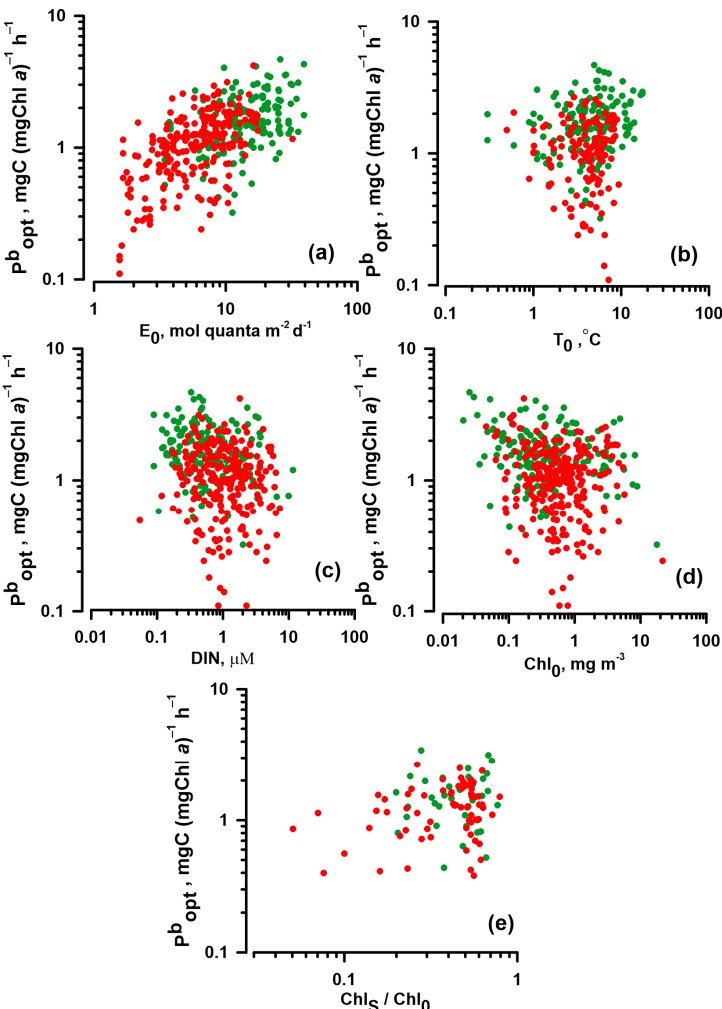

**Figure 3.** The relationships between the optimal assimilation number ($P^b_{opt}$) and environmental factors. (**a**) Subsurface PAR ($E_0$); (**b**) sea surface temperature ($T_0$); (**c**) concentration of dissolved inorganic nitrogen (DIN); (**d**) surface chlorophyll *a* concentration ($Chl_0$); and (**e**) the ratio of chlorophyll *a* concentration of small (<3 μm) to total surface phytoplankton ($Chl_s/Chl_0$). Green and red colours indicate the measurements performed in the summer and autumn, respectively.

The relationships between different abiotic factors are characterised by multicollinearity (Table 5), which leads to uncertainty in the estimations of its influence on $P^b_{opt}$. Principal component analysis (PCA) allows a reduction in the multicollinearity effect. PCA also generates an ordination diagram that illustrates links between $P^b_{opt}$ and environmental factors.

**Table 5.** The correlation matrix between the log-transformed values of environmental variables. R—coefficient of correlation; *p* value—statistical significance of R; *N*—the number of data. $T_0$—surface water temperature; $S_0$—surface salinity; $PO_4$, $Si(OH)_4$ and DIN—surface concentrations of phosphates, dissolved silicon, and dissolved inorganic nitrogen, respectively; $Chl_0$—surface chlorophyll *a* concentration; $E_0$—subsurface PAR. The asterisks indicate significant correlations ($p < 0.05$).

| Parameter | Statistics | $T_0$ | $S_0$ | $Chl_0$ | $E_0$ | $PO_4$ | $Si(OH)_4$ | DIN |
|---|---|---|---|---|---|---|---|---|
| T | R | 1.00 | | | | | | |
| | N | 411 | | | | | | |
| | p | $<10^{-3}$ | | | | | | |
| $S_0$ | R | −0.22 * | 1.00 | | | | | |
| | N | 408 | 408 | | | | | |
| | p | $<10^{-3}$ | $<10^{-3}$ | | | | | |
| $Chl_0$ | R | 0.30 * | −0.61 * | 1.00 | | | | |
| | N | 411 | 408 | 411 | | | | |
| | p | $<10^{-3}$ | $<10^{-2}$ | $<10^{-3}$ | | | | |
| $E_0$ | R | 0.11 * | 0.07 | −0.20 * | 1.00 | | | |
| | N | 395 | 392 | 395 | 395 | | | |
| | p | 0.036 | 0.163 | $<10^{-3}$ | $<10^{-3}$ | | | |
| $PO_4$ | R | −0.14 * | −0.21 * | 0.19 * | −0.06 | 1.00 | | |
| | N | 404 | 401 | 404 | 388 | 404 | | |
| | p | 0.006 | $<10^{-3}$ | $<10^{-3}$ | 0.232 | $<10^{-3}$ | | |
| $Si(OH)_4$ | R | 0.21 * | −0.52 * | 0.67 * | −0.04 | 0.35 * | 1.00 | |
| | N | 410 | 407 | 410 | 394 | 404 | 410 | |
| | p | $<10^{-3}$ | $<10^{-2}$ | $<10^{-2}$ | 0.473 | $<10^{-3}$ | $<10^{-3}$ | |
| DIN | R | −0.11 * | −0.20 * | 0.28 * | −0.23 * | 0.28 * | 0.25 * | 1.00 |
| | N | 387 | 384 | 387 | 377 | 385 | 387 | 387 |
| | p | 0.027 | $<10^{-3}$ | $<10^{-3}$ | $<10^{-3}$ | $<10^{-3}$ | $<10^{-3}$ | $<10^{-3}$ |

The results of PCA, which are presented in Figure 4, suggest that the main variables that contribute to the first principal component (PC1) are the abiotic factors $S_0$, $Si(OH)_4$, and $Chl_0$. PC1 describes 29.4% of the total variance. The second principal component (PC2) includes the main variables of $P^b_{opt}$ and $E_0$ and describes 21.8% of the total variance (Figure 4a).

The PCA analysis illustrates the well-pronounced positive relation between $P^b_{opt}$ and $E_0$ that is indicated by the same direction of their vectors on the factorial plane. Similar to the correlation analysis, the results of the PCA show that $P^b_{opt}$ is weakly linked with salinity and $Si(OH)_4$ as indicators of riverine waters. This is indicated by their orthogonality on the PCA map. Furthermore, the weak relationships between $P^b_{opt}$ and surface temperature and nutrients were shown by the PCA results (Figure 4a).

The contribution of PC1 and PC2 to the individual samples is shown on the factorial plane (Figure 4b). For visibility, all samples were divided according to the range in salinity. Figure 4b shows that the individual samples collected at different salinity were strongly divided. Samples at $S_0 < 25$ and $S_0 > 25$ were, respectively, positively and negatively influenced by PC1, while the influence of PC2 was rather equal. This finding suggests that there are no differences between the links of $P^b_{opt}$ with $E_0$ across the salinity gradient.

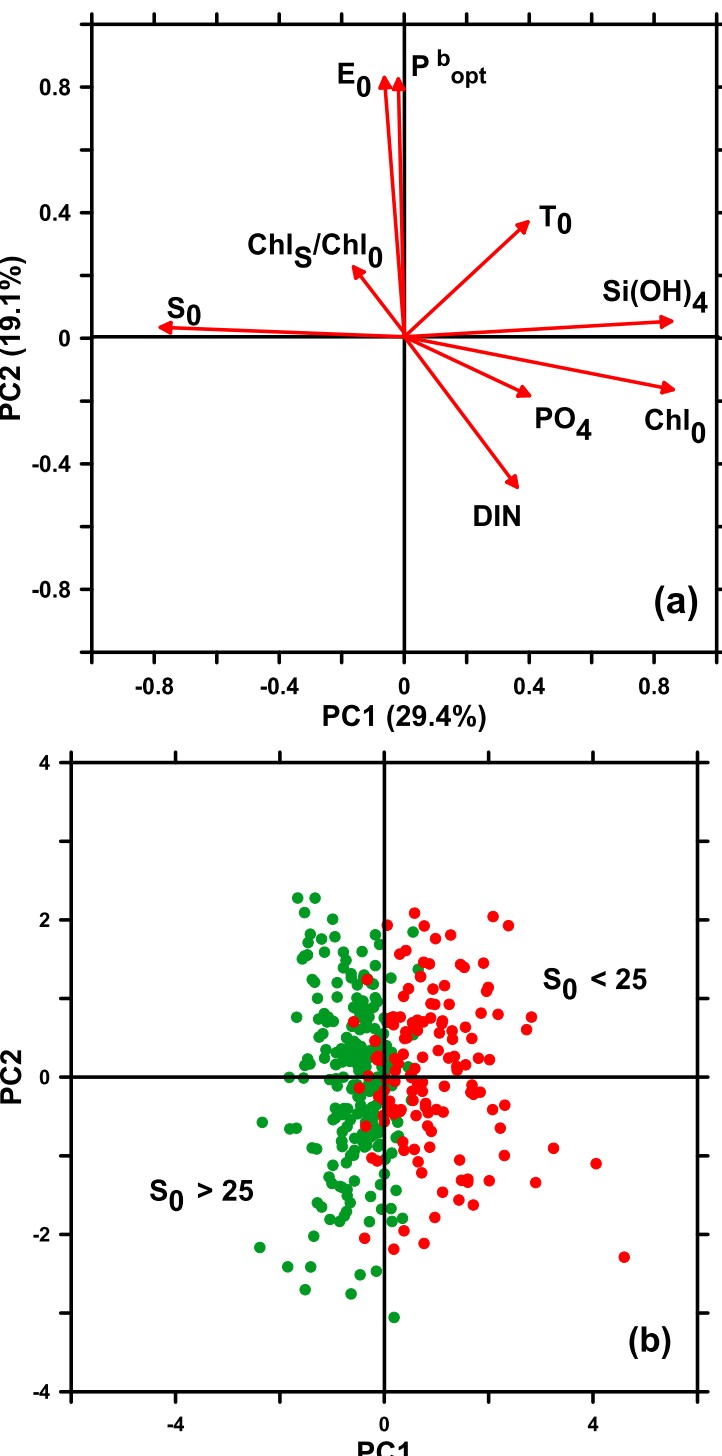

**Figure 4.** Graphical representation of the results of the principal component analysis (PCA). (**a**) Correlations on the factorial plane formed with the two first principal components (PC1 and PC2). The values at the axis designate the PC1 and PC2 contribution to the total variance. The values of the optimal assimilation number ($P^b_{opt}$), surface chlorophyll *a* concentration ($Chl_0$), subsurface PAR ($E_0$), surface water temperature ($T_0$), surface salinity ($S_0$), phosphate ($PO_4$), dissolved silicon ($Si(OH)_4$), dissolved inorganic nitrogen (DIN), and the ratio of chlorophyll *a* concentration of small (<3 μm) to total surface phytoplankton ($Chl_s/Chl_0$) are presented. (**b**) Projection of the individual samples collected in the brackish ($S_0 < 25$) (red colour) and oceanic ($S_0 > 25$) (green colour) waters of the Siberian Seas on the factorial plane.

### 3.3. $P^b_{opt}$ Model Developed with $E_0$, Its Efficiency, and a Comparison with T-Based Models

It was mentioned above that a strong correlation was obtained between $P^b_{opt}$ and $E_0$ (Table 4). There was no other abiotic parameter that was related to $P^b_{opt}$ so closely. Thus, it is reasonable to use $E_0$ as the only abiotic factor in the $P^b_{opt}$ region-specific regression model ($E_{0reg}$). To develop this model, we used two-thirds of the dataset as mentioned in [50]. The equation of liner regression relating log-transformed values of $P^b_{opt}$ and $E_0$ is

$$\log_{10} P^b_{opt} = 0.537 \log_{10} E_0 - 0.399 \ (R = 0.62, N = 266) \tag{1}$$

The results of $E_{0reg}$ verification using field observations are represented in Table 6 and Figure 5a. The comparison of the measured and modelled values of $P^b_{opt}$ suggests that $E_{0reg}$ overestimates the field data on average (the average absolute error (B) is equal to 0.040). The root-mean-square difference (RMSD) value implies that the calculated values of $P^b_{opt}$ were 1.7-fold higher than the field data on average.

**Table 6.** Regression statistics and performance indices for the log-transformed measured and modelled optimal assimilation number ($P^b_{opt}$). Slope and intercept are parameters of the linear regressions; $R^2$—coefficient of determination; *N*—number of data used for model validation; *p*-value indicates the significance level of each regression. Indices are the mean model bias (B), the standard deviation of the log-transformed modelled values of $P^b_{opt}$ ($\sigma$), and the root-mean-square difference (RMSD); $E_{0reg}$—the region-specific $P^b_{opt}$ model developed using subsurface photosynthetically available radiation (PAR) and verified with field data. BF-model—the $P^b_{opt}$ model developed by Behrenfeld and Falkowski [13]. $T_{reg}$—the region-specific $P^b_{opt}$ model developed using the relationship between $P^b_{opt}$ and sea surface temperature. $E_{sat}$—$E_{0reg}$ verified with satellite-derived data of PAR.

| Model | Regression Statistics | | | | | Performance Indices | | |
|---|---|---|---|---|---|---|---|---|
| | Slope | Intercept | $R^2$ | *N* | *p* Value | B | $\sigma$ | RMSD |
| $E_{0reg}$ | 0.697 | 0.879 | 0.35 | 131 | <0.05 | 0.040 | 0.176 | 0.227 |
| BF-model | 0.060 | 0.405 | 0.02 | 410 | <0.05 | 0.343 | 0.409 | 0.444 |
| $T_{reg}$ | 0.036 | 0.115 | 0.03 | 137 | <0.05 | 0.066 | 0.055 | 0.279 |
| $E_{sat}$ | 0.289 | 0.226 | 0.19 | 373 | <0.05 | 0.183 | 0.190 | 0.321 |

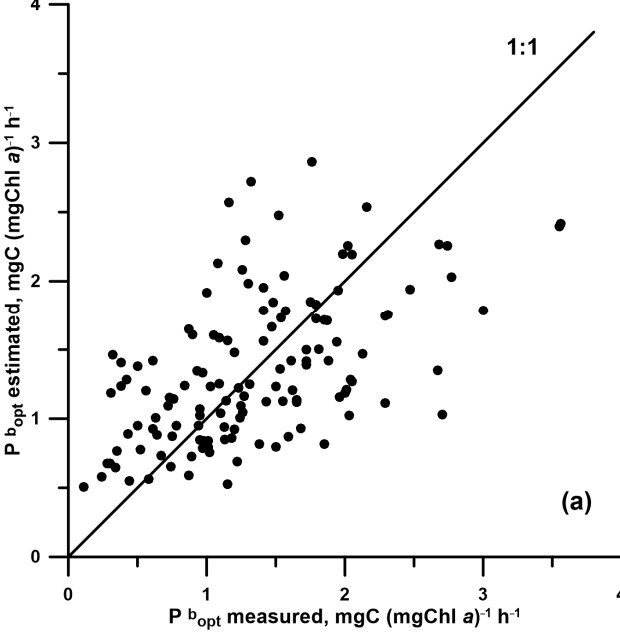

**Figure 5.** *Cont.*

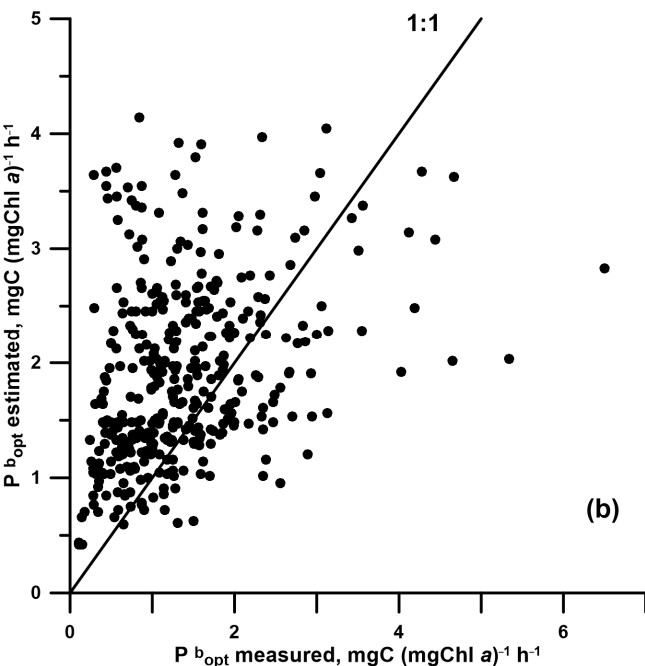

**Figure 5.** (**a**) A comparison of the values of the optimal assimilation number ($P^b_{opt}$) measured and calculated using subsurface PAR ($E_0$) and (**b**) a comparison of the values of $P^b_{opt}$ measured and calculated using satellite-derived PAR ($E_{sat}$) (black points). The solid line indicates 1:1 correlation.

The application of the developed algorithm to study the spatiotemporal variations in $P^b_{opt}$ and for water column primary production (IPP) estimations assumes the introduction of satellite-derived PAR ($E_{sat}$) into Equation (1). Therefore, it is appropriate to validate the $E_{0reg}$ model using satellite-derived data. The results represented in Table 6 and Figure 5b suggest that the application of $E_{sat}$ decreases the model performance by a factor of 1.4 according to the RMSD value. The correlation between the measured and modelled values of $P^b_{opt}$ decreased by a factor of 1.8 in comparison with the data of verification using field observations ($R^2 = 0.19$ and 0.35, respectively) (Table 6). Furthermore, the introduction of $E_{sat}$ into Equation (1) enhanced B by a factor of 4.6.

In the models used for IPP estimation using satellite-derived data, often $P^b_{opt}$ is retrieved with the polynomial function derived using the worldwide dataset (Equation (11) in [13], the BF model in further) or with the regional-adopted relationships between $P^b_{opt}$ and $T_0$ [25,78,79]. In that regard, it is useful to compare the model performances of the BF model and $E_{0reg}$ for the estimation of their skill in the SSs. Figure 6a presents the distribution of the $P^b_{opt}$ dataset related to $T_0$ and the curve of the polynomial function that links $P^b_{opt}$ and $T_0$ from [13]. The result presented in Figure 6a implies that the BF model will dramatically overestimate $P^b_{opt}$ in the SSs. This conclusion is confirmed with the results of the verification of the BF model using the $T_0$ dataset collected in the SSs (Table 6, Figure 6b). The value of B characterising the error in the BF model is equal to 0.343, which is 9.5-fold higher than that of $E_{0reg}$. The coefficient of determination ($R^2$) of the BF model is 17-fold lower than that of $E_{0reg}$ (0.35 and 0.02, respectively). The value of RMSD of the BF model is 1.9-fold higher than that of $E_{0reg}$ (Table 6). The index of efficiency of the BF model (RMSD = 0.444) suggests that $P^b_{opt}$ values calculated using this algorithm can over- or underestimate the measured ones by a factor of 2.8, which is 1.6-fold higher than in the case of $E_{0reg}$ application.

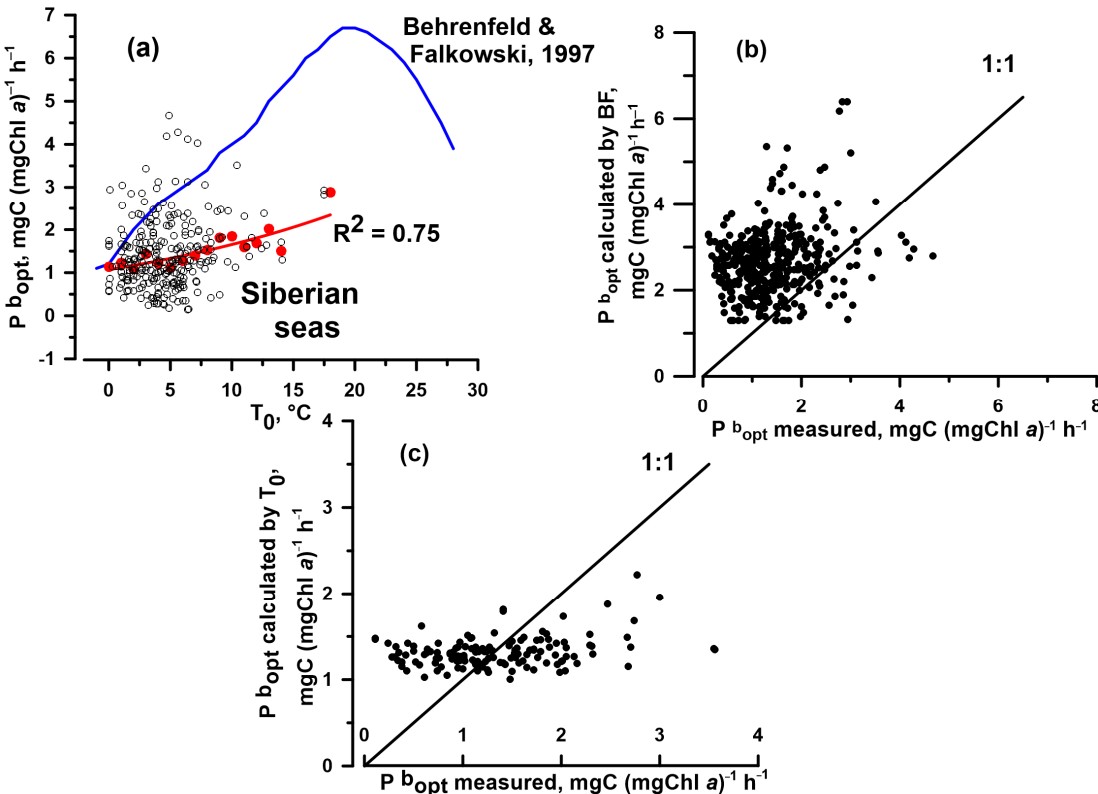

**Figure 6.** (**a**) the optimal assimilation number ($P^b_{opt}$) vs. sea surface temperature ($T_0$) in the Siberian Seas (SSs) (open circles) in comparison with polynomial regression obtained by Behrenfeld and Falkowski [13] (BF-model) based on the worldwide dataset (the blue line). Red colour indicates the exponential relationship between $P^b_{opt}$ and $T_0$ obtained based on the SSs dataset (T_reg—algorithm). (**b**) A comparison of the values of $P^b_{opt}$ measured and calculated using the BF-model. (**c**) A comparison of the values of $P^b_{opt}$ measured and calculated using T_reg. The solid line indicates a 1:1 correlation.

To illustrate the problems connected with the estimation of $P^b_{opt}$ in the SSs using $T_0$ solely, the authors developed the region-specific empirical algorithm based on the relationship between $P^b_{opt}$ and $T_0$ established using the SSs dataset (*N* = 266) (T_reg). In the development of this model, the authors followed the approach described in [13]. The median values of $P^b_{opt}$ were calculated for each 1 °C temperature span in the range from 0 to 18 °C. The relationship between the median values of $P^b_{opt}$ and $T_0$ was described using the exponential function (Figure 6a):

$$P^b_{opt} = 1.07 \, e^{0.044 \, T_0}. \tag{2}$$

This model was validated with the independent dataset (data that were not used for model development). The results of this verification are presented in Table 6 and Figure 6c. As in the case of applying the BF model, weak links between the measured and modelled values of $P^b_{opt}$ ($R^2$ = 0.03) were observed. The value of RMSD that was equal to 0.279 suggests that the modelled values of $P^b_{opt}$ can 1.9-fold over- or underestimate the measured ones. The average absolute error of T_reg was equal to 0.066, which was 1.8-fold higher than in the case of $E_{0reg}$ application. Thus, it can be concluded that the T_reg algorithm is not applicable for $P^b_{opt}$ estimation in the SSs.

### 3.4. The Spatial Distribution in $P^b_{opt}$ Assessed Using Satellite-Derived Data

The introduction of $E_{sat}$ data into Equation (1) allows retrieving the pattern of the spatial distribution in $P^b_{opt}$ over the entire area of the SSs. In Figure 7, satellite climatolo-

gies (2007–2020) of $P^b_{opt}$ from July to October are presented. It should be noted that for averaging, the years were chosen that coincided with the field observations (Table 1). As expected, the spatial distribution of $P^b_{opt}$ was quasi-latitudinal and follows by the spatial distribution in PAR. The values of $P^b_{opt}$ basically decreased northward (Figure 7).

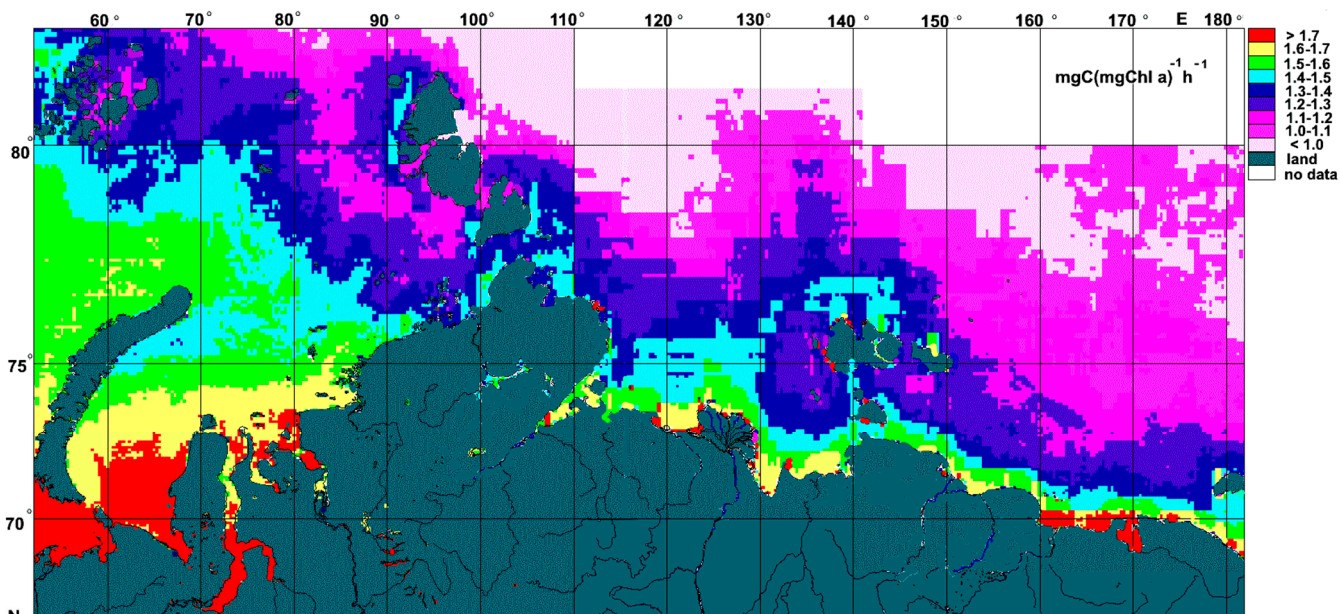

**Figure 7.** The representation of Siberian Seas estimates of the optimal assimilation number ($P^b_{opt}$) from satellite climatologies (2007–2020) calculated using $E_{0reg}$ model for the period from July to October.

## 4. Discussion

### 4.1. The Average Values and Spatiotemporal Variations in the Optimal Assimilation Number ($P^b_{opt}$) in the Siberian Seas (SSs)

The average value of $P^b_{opt}$ in the SSs (1.38 mgC (mg Chl $a^{-1}$ $h^{-1}$) was in the range of variability observed in the north of Baffin Bay (0.3–4.1 mgC (mg Chl $a^{-1}$ $h^{-1}$) [80]. In addition, it was close to the values measured in the Chukchi and Beaufort Seas (from 0.6 to 1 mgC (mg Chl $a^{-1}$ $h^{-1}$, on average) [81], and it was higher than historical (1956, 1961–1963) values observed in the Canadian Arctic (0.2–0.4 mgC (mg Chl $a^{-1}$ $h^{-1}$) [82]. It is known that an accurate as possible calculation of the average value of $P^b_{opt}$ within a particular biogeochemical province [48] is critically important for the estimation of water column primary production (IPP) [21].

The average values of $P^b_{opt}$ for small (<3 μm) phytoplankton were higher than those for large (>3 μm) phytoplankton in the entire SS (Table 2). This finding is consistent with the investigations that established that the chlorophyll-specific carbon fixation rate declined with a decrease in cell sizes [83–87]. Theoretically, the specific photosynthetic rate of small phytoplankton must be higher than that of the large fraction because of the high size-to-volume ratio that allow it to be more effective at absorbing light and nutrients. As a result, small cells have advantages in conditions of low nutrients and irradiance [34,88,89]. This assumption is confirmed with the analysis of extensive field and laboratory data [90].

The river runoff on the Siberian shelf controversially influences primary production (PP) characteristics. On the one hand, a large amount of dissolved (DOM) and particulate (POM) organic matter of river genesis limits the photosynthetic rate in the water column by decreasing water transparency and euphotic depth as a consequence [47]. On the other hand, large rivers enrich the coastal areas of the SSs with nutrients [37,38], increasing the photosynthetic capacity of phytoplankton. Thus, it can be assumed the differences in the values of $P^b_{opt}$ between the river runoff regions and those where the influence of rivers is insignificant. Nevertheless, there were no statistically significant differences between

the average values of $P^b_{opt}$ in brackish, with surface salinity ($S_0$) < 25, and oceanic waters ($S_0$ > 25). Thus, it can be concluded that $P^b_{opt}$ in the SSs is influenced by river runoff to a small extent.

To evaluate the relationships between $P^b_{opt}$ and environmental factors in waters with different salinity, the dataset was differentiated according to the $S_0$ values (Table 7). The results of the correlation analysis suggest that the links between $P^b_{opt}$ and $E_0$ were not significantly different in brackish and oceanic waters (R = 0.59 and 0.63, respectively). Thus, these findings suggest that the developed $P^b_{opt}$ algorithm can be used as universal both in the river runoff regions of the SSs and in the areas out of such influence.

**Table 7.** The correlation matrix between the log-transformed optimal assimilation number and environmental variables in the areas with $S_0$ < 25 and $S_0$ > 25. $P^b_{opt}$—optimal assimilation number of the total phytoplankton; R—coefficient of correlation; *p*-value—statistical significance of R; *N*—the number of data; $T_0$—sea surface temperature; $S_0$—sea surface salinity; $PO_4$, $Si(OH)_4$, and DIN—surface concentrations of phosphates, dissolved silicon, and dissolved inorganic nitrogen, respectively; $Chl_0$—chlorophyll a concentration of the surface total phytoplankton; $E_0$—subsurface photosynthetically available radiation. The asterisks indicate significant correlations (*p* < 0.05).

| Parameter | Statistics | $T_0$ | $S_0$ | $Chl_0$ | $E_0$ | $PO_4$ | $Si(OH)_4$ | DIN |
|---|---|---|---|---|---|---|---|---|
| $P^b_{opt}$ $S_0$ < 25 | R | 0.24 * | 0.21 * | −0.10 | 0.59 * | −0.09 | 0.19 * | −0.24 * |
| | N | 159 | 156 | 159 | 150 | 154 | 158 | 147 |
| | p | 0.003 | 0.009 | 0.211 | <$10^{-3}$ | 0.284 | 0.020 | 0.003 |
| $P^b_{opt}$ $S_0$ > 25 | R | 0.02 | −0.11 | −0.41 * | 0.63 * | 0.02 | 0.06 | −0.19 * |
| | N | 245 | 254 | 254 | 247 | 252 | 254 | 242 |
| | p | 0.713 | 0.075 | <$10^{-3}$ | <$10^{-2}$ | 0.696 | 0.376 | 0.002 |

A decrease in $P^b_{opt}$ from July to October is explained by a decline in the values of the main environmental factors, generally subsurface photosynthetically available radiation (PAR) (Figure 2). Similarly, the tendency toward a decrease in $P^b_{opt}$ at the end of the growing season was noted in other regions of the Arctic Ocean [80].

### 4.2. Influence of Environmental Factors on $P^b_{opt}$

In theory, the relationships between $P^b_{opt}$, as well as the maximal assimilation number ($P^b_{max}$), and the main environmental factors must be the same. Therefore, in this section, it is meaningful to discuss the relationships between environmental factors and both $P^b_{opt}$ and $P^b_{max}$ due to the most representation of the latter.

#### 4.2.1. Influence of PAR on $P^b_{opt}$

The results obtained in this study allow us to characterise the SSs phytoplankton, on the one hand, as highly photoadaptive and, on the other hand, as light-limited. The strong correlation between $P^b_{opt}$ and PAR is evidence that day-to-day variations in incident radiation have a fast influence on changes in carbon fixation rate. Thus, the SS phytoplankton is capable of fast photoadaptation. In earlier studies, it was shown that in the Arctic Ocean, the assimilation number ($P^b$) linearly and positively depended on PAR [47,53,91,92]. The absence of a "plateau" on the curves of the relationships between $P^b$ and PAR implies that arctic phytoplankton is light-limited. For that reason, the values of $P^b_{opt}$ are usually registered under saturated irradiance.

Often, the values of $P^b_{opt}$ are observed within the upper mixed layer (UML). Therefore, some authors considered the links of $P^b_{opt}$ with the average values of PAR in the UML ($E_{UML}$) [50]. The findings of the present study suggest that in the SSs, $P^b_{opt}$ was better correlated with subsurface PAR ($E_0$) (R = 0.61, *p* < 0.01, N = 397) than with $E_{UML}$ (R = 0.54, *p* < 0.01, N = 397). This result can be explained by the fact that in 97% of cases, the highest

values of $P^b_{opt}$ were observed in the subsurface layer of 0–2 m ($P^b_0$), and a strong positive correlation was established between the log-transformed values of $P^b_{opt}$ and $P^b_0$ (Figure 8). Observation of $P^b_{opt}$ predominantly in the subsurface layer suggests a more pronounced light limitation of the photosynthetic rate in the SSs in comparison with other regions of the Arctic Ocean where $P^b_{opt}$ can be registered at the depths of the deep maxima of chlorophyll and PP [81]. This phenomenon is linked with the optically complex type of waters in the SSs [93] enriched by DOM and POM of river genesis [37,94–96].

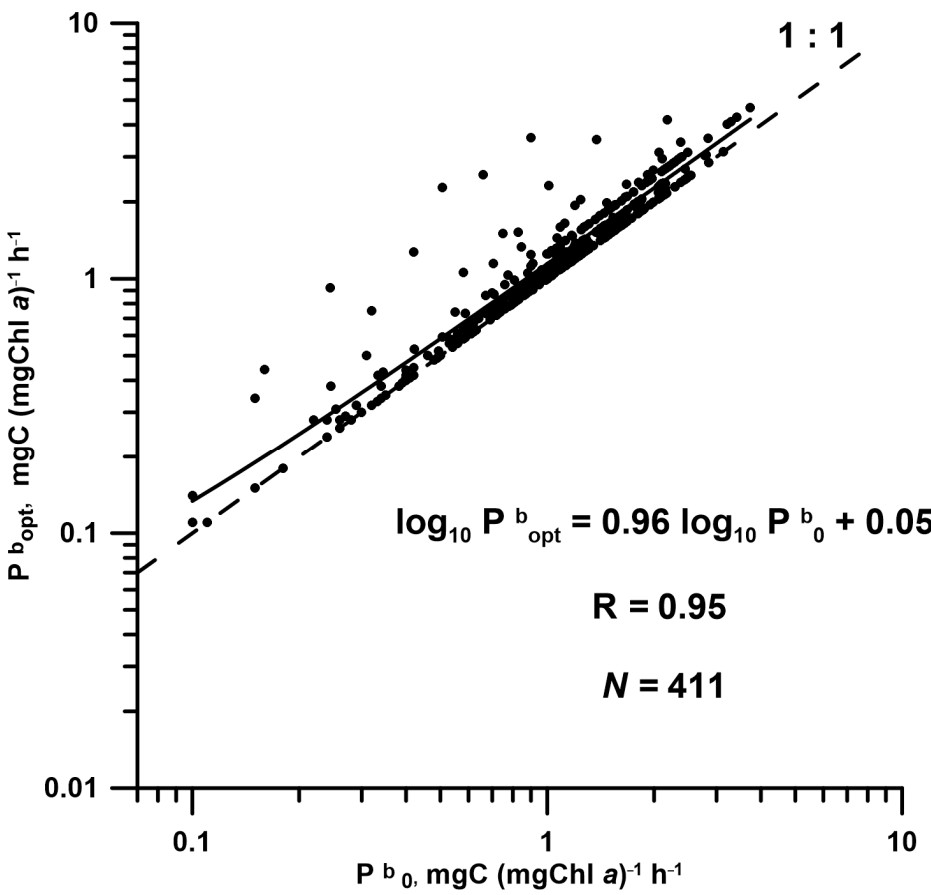

**Figure 8.** The relationship between the optimal assimilation number ($P^b_{opt}$) and surface assimilation number ($P^b_0$) (black points) in the Siberian Seas. The solid line shows the line of regression. The dashed line indicates 1:1 correlation.

### 4.2.2. Influence of Temperature and Nutrients on $P^b_{opt}$

Generally, a weak correlation has been noted in the Arctic Ocean between $P^b$ and temperature [97–100]. Furthermore, it was established by [101] that the photosynthetic parameters in the Arctic Ocean were not influenced by temperature over the range from −2 to 8 °C. In the dataset used in the presented study, 90% of the values of $T_0$ fit in this diapason (Table S1). Thus, our findings are consistent with the outcomes of the previous studies.

The absence of a close relationship between $P^b_{opt}$ and $T_0$ was explained simultaneously by a negative correlation between $T_0$ and nutrients and a positive correlation between $T_0$ and $E_0$ [83,102]. In accordance with those findings, the authors registered in the SSs significant, but weak, relationships between $T_0$ and the concentration of dissolved inorganic nitrogen (DIN) and between $T_0$ and $E_0$ (Table 5). This result can be explained by a mismatch of seasonal maxima in $T_0$, DIN, and $E_0$ in the SSs. High values of $T_0$ and $E_0$ are observed in July and August when nutrients are exhausted [103].

The role of nutrients in PP in the Arctic Ocean is well known [51,52,104]. On the other hand, it is difficult to establish close links between PP characteristics and nutrient concentration [32,47,98,105]. There are many reasons for that. A low nutrient concentration very often is not evidence of a low increment in phytoplankton biomass due to grazing by zooplankton, as well as cell death and sedimentation [106]. Moreover, the enrichment in UML by nutrients during the winter convection usually does not coincide with the high values of $P^b$ registered during the phytoplankton bloom in spring. Furthermore, in conditions of high nutrients, the photosynthetic rate can decrease because of energetic competition between the DIN assimilation process and the Calvin cycle [107]. On the other hand, during the limitation of nutrients, phytoplankton can use dissolved organic nitrogen for growth and photosynthesis [108,109] and keep a relatively high photosynthetic rate.

All the reasons listed above can lead to unpredictable, positive or negative, relationships between productivity parameters and nutrients. In the present study, a statistically significant but weak correlation was found only between $P^b_{opt}$ and DIN (Table 4). The negative relationship between $P^b_{opt}$ and DIN is determined by the spatial distribution in surface nutrients in the SSs. Thus, the main source of nutrients in the inner shelf of the SSs is the river discharge [37,38]. In these areas, an increase in nutrient concentration is accompanied by a high Chl *a* content. In turn, the negative correlation between $P^b_{opt}$ and Chl *a* (R = –0.21, $p < 10^{-3}$, N = 411) (Table 4) to a high extent determines the opposite link between $P^b_{opt}$ and DIN.

### 4.2.3. Modelling of $P^b_{opt}$ and Its Application for Remote Sensing

The results of the correlation analysis and PCA suggest that light is the main environmental factor that constrains the assimilation activity of phytoplankton in the SSs (Figure 4, Table 4). The variability in $E_0$ explained 37% of $P^b_{opt}$ variations ($R^2 = 0.37$). Therefore, it can be assumed that $E_0$ can be used as a single input variable to the empirical model of $P^b_{opt}$.

A weak correlation between $P^b_{opt}$ and $T_0$ does not allow for using the latter parameter as a predictor of the assimilation activity of phytoplankton in the SSs. In the present article, it was revealed that the application of the dependence between $P^b_{opt}$ and $T_0$ for the World Ocean [13] led to a significant error in the calculations of $P^b_{opt}$ in the SSs (Table 6). Using the region-specific relationship between $P^b_{opt}$ and $T_0$ did not improve the predictive capacity of the temperature-based model. Thus, according to the authors' dataset, the empirical formula (1) is the best approximation of $P^b_{opt}$, and it can be used for the calculation of this parameter using satellite-derived $E_0$ ($E_{sat}$).

As was mentioned above, using $E_{sat}$ as an input variable decreased the efficiency of the developed model (1). This result was connected with the errors in $E_{sat}$ determination [77,110]. To estimate this error, a comparison of the field data of $E_0$ with matched-up in space and time values of $E_{sat}$ obtained by a MODIS-Aqua scanner was carried out. The results of this comparison suggest that the values of $E_{sat}$ overestimate the field observations (Figure 9). As a consequence, the calculations of $P^b_{opt}$ using $E_{sat}$ also were overestimated in comparison with the field data as evidenced by the positive bias of linear regression (Table 6).

Another approach to the estimation of $P^b_{opt}$ is the application of $T_0$ and $Chl_0$, also registered using a satellite scanner, as additional input variables in the model [50]. To verify whether model predictive capacity improves after using $T_0$ and $Chl_0$ together with $E_0$, the equation of multiple regression linking the log-transformed values of these variables with $P^b_{opt}$ was obtained as:

$$\log_{10} P^b_{opt} = 0.512 \log_{10} E_0 + 0.015 \log_{10} T_0 - 0.070 \log_{10} Chl_0 - 0.408 \ (R = 0.64, N = 266). \tag{3}$$

The verification of the developed model (3) suggests that the input of $T_0$ and $Chl_0$ to the calculations does not improve the model skill in comparison with $E_{0reg}$ (Table 6). The main parameters of model efficiency were as follows: $R^2 = 0.34$, RMSD = 0.228. Thus, according to the authors' findings, it is sufficient to use only $E_0$ in the $P^b_{opt}$ model as the most relevant abiotic parameter.

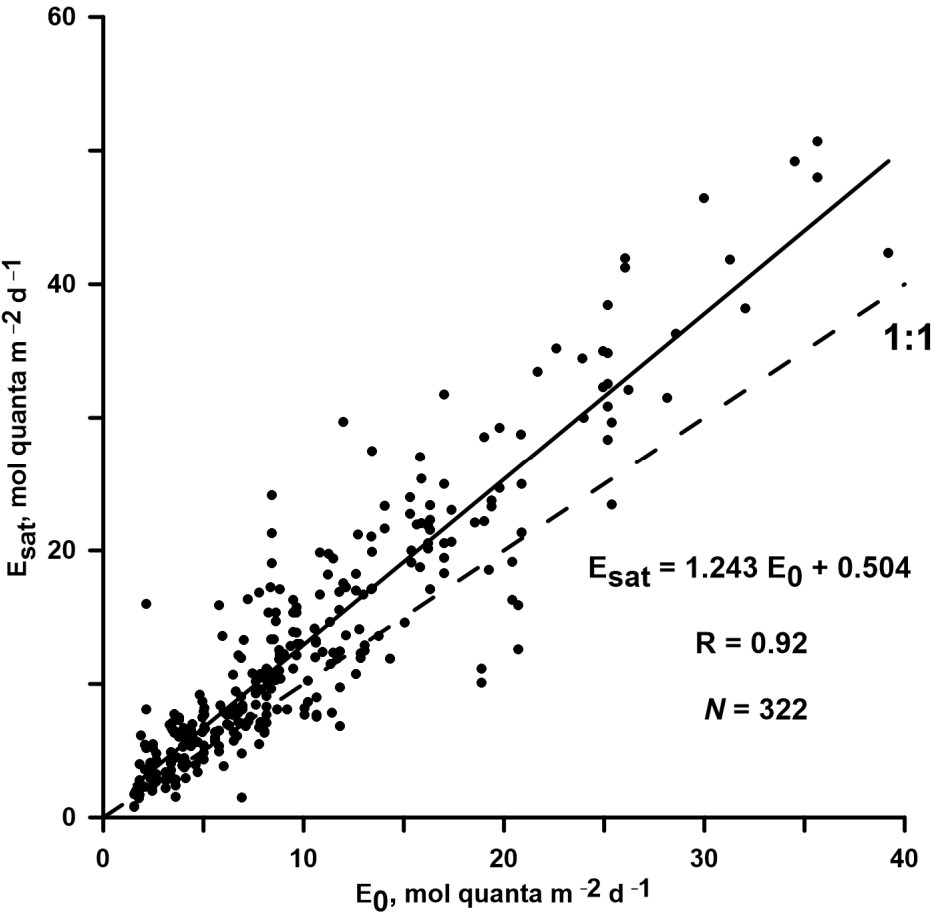

**Figure 9.** The relationship between measured ($E_0$) and satellite-derived subsurface PAR ($E_{sat}$) (black points). The solid line shows the line of regression. The dashed line indicates a 1:1 correlation.

The tendency of the latitudinal distribution in $P^b_{opt}$ in the SSs obtained using the Equation (1) with $E_{sat}$ as an input variable (Figure 7) was consistent with the global distribution in $P^b_m$ according to [50]. In that study, where the spatial distribution in the assimilation number was assessed using PAR, temperature, and chlorophyll *a* concentration, the values of $P^b_m$ also decreased poleward (Figure 8 in [50]).

## 5. Conclusions

One objective of the present study was to develop a simple algorithm for $P^b_{opt}$ estimation that would be useful to evaluate in future IPP in the Siberian Seas (SSs) using chlorophyll-based models and satellite observations. For this purpose, it was needed to choose an environmental parameter that would be most closely linked with $P^b_{opt}$ and easily detected from space. The analysis of the dataset used in this article suggests that incident PAR is the required variable because its role in the variability in $P^b_{opt}$ is dominant.

In the present article, it is shown that the commonly used algorithm for the calculation of the optimal assimilation number ($P^b_{opt}$) based on sea surface temperature ($T_0$) registered using a satellite is badly applicable for phytoplankton of high latitudes, in particular, in the SSs. As a consequence, the application of the dependence linking $P^b_{opt}$ with $T_0$ must lead to errors in the estimation of the annual value of water column primary production (IPP) using chlorophyll-based models and satellite-derived data. The findings of this study suggest that the application of photosynthetically available radiation can be sufficient for the adequate estimation of $P^b_{opt}$ in light-limited regions. The main practical result of this study is the developed empirical region-specific algorithm of $P^b_{opt}$, which can be used in the future for IPP estimation in the Arctic Ocean.

**Supplementary Materials:** The following supporting information can be downloaded at: https://www.mdpi.com/article/10.3390/jmse11030522/s1, Table S1: Phytoplankton productivity parameters and environmental variables in the Kara Sea from 1993 to 2020; Table S2: Matched-up points of in situ and satellite-derived assimilation numbers; Table S3. Matched-up points of in situ and satellite-derived subsurface PAR; Figure S1. Frequency distribution of log-transformed values of biotic and abiotic variables. (a)–optimal assimilation number ($P^b_{opt}$); (b)–surface chlorophyll *a* concentration ($Chl_0$); (c)–subsurface photosynthetically available radiation (PAR) ($E_0$); (d)–sea surface temperature ($T_0$); (e)–sea surface salinity ($S_0$); (f)–concentration of dissolved inorganic nitrogen (DIN); (g)–concentration of phosphates ($PO_4$); (h) –concentration of dissolved silicon ($Si(OH)_4$); (i)–the ratio of chlorophyll *a* concentration of small (<3 μm) to total surface phytoplankton ($Chl_s/Chl_0$); Kα–value of Kolmogorov-Smirnov test; *p* value–statistical reliability; *N*–number of data. Solid line is the curve of expected normal distribution.

**Author Contributions:** A.B.D. is the author who was responsible for the conception of this article and the main contributor to the text. T.A.B. and S.V.S. contributed to the collection and treatment of the data. All authors have read and agreed to the published version of the manuscript.

**Funding:** This study was supported by the Russian Science Foundation (project No. 23-27-00061, the scientific direction is "Phytoplankton size structure of the Kara Sea: environmental control of ecophysiological parameters and significance in estimation of primary production").

**Institutional Review Board Statement:** Not applicable.

**Informed Consent Statement:** Not applicable.

**Data Availability Statement:** Raw field data are appended in the Supplementary Materials. MODIS-Aqua data used were obtained from the NASA website (http://oceancolor.gsfc.nasa.gov (accessed on 22 August 2022) the Goddard Distributed Active Archive Center under the auspices of the National Aeronautics and Space Administration.

**Acknowledgments:** We are grateful to Polukhin A.A. for kindly providing of hydrochemical data and to Gagarin V.I. (Shirshov Institute of Oceanology, Russian Academy of Sciences) for help in the treatment of satellite data.

**Conflicts of Interest:** The authors declare no conflict of interest.

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
