# Peer review of "Optimal Assimilation Number of Phytoplankton in the Siberian Seas: Spatiotemporal Variability, Environmental Control and Estimation Using a Region-Specific Model"

_jmse, doi:10.3390/jmse11030522_

Round 1
Reviewer 1 Report
The manuscript (MS) is about the optimal assimilation number of phytoplankton in SS and discusses its spatiotemporal variability, relations with environmental variables and the discussion about the use of remote sensing data Esat more specifically to predict Pbopt using a proposed model.
Saying this I believe that the title chosen is not adequate. In the MS there isn't an estimation made by remote sensing. There are no maps or graphs with PP or Chla estimations from satellite data. for comparison with the field surface data for example. What is done is the use of Esat to the model presented and its comparison with field data. The title includes the phrase "environmental control and estimation by satellite remote sensing" which I think is not a good choice. I recommend that you change the title according to what you have in the MS which is the adequacy of remote sensing data to the prediction of Pbopt in SS by models. Otherwise, you will need to put in the MS a point /paragraph regarding remote sensing data for the studied period.
I consider that the MS is well-written and structured and that it supports the point that regional models have to be developed rather than using one all-purpose model worldwide. This is the strongest merit of this MS, and that´s another reason I suggest that the title is changed. Besides it doesn't matches with the abstract provided.
Specific considerations
Tables 2 and 3 lack the units of the measurements please add that to the tables themselves or in the caption to help the reader. It took me a little to understand what those numbers referred to.
Figure 3 - standard deviation bars overlap between variables and are difficult to understand which belongs to whom. Use colour or texture for the bars.
Page 8 point 3.2 - Correlations coefficients below R=0.5 even if are statistically significant do not mean a thing environmentally speaking. This is what I call "trying to stretch a rope". Remove this. I understand that you've mentioned this here because of points 4.2.2 to 4.2.4 which I'll refute up front.
Table 4 - column Chls/Chl0, the p-value for PboptL is 0.256 and you´ve marked that coefficient of correlation as significative. 0.256>0.05. Verify the value and correct all that it related to this.
Page 10, paragraph 2, line 6 - where is "Figure 3b" don´t you mean 5b? there is no 3b in the MS. Correct this
Page 12, paragraph 1, last phrase. "using two-thirds". When I first read this I thought why the use of only two-thirds? then I read the entire MS to find where this is mentioned again. I think it would be better to add a few words like "as mentioned in ..." for the reader to easily understand where you are at this phase of the MS regarding modelling.
Page 18, discussion of points 4.2.2, 4.2.3, 4.2.4. - The discussion made around correlation coefficients less than 0.5 is very doubtful and realising that explains less than even 10% of the data is meaningless. That is just an empirical mathematical discussion with no real impact on the environmental discussion. Remove this.
Unless you can prove that it can explain a specific process/condition that will need to be addressed or taken into account in the future, for model prediction in the SS region and that it might help to understand the impact of climate change or environmental changes due to pollution, do rewrite the paragraphs otherwise it is not important and adds clutter to the discussion.
Page 18, point 4.2.5, paragraph 1, line 3 - "E0 explained 37%". Where does this value come from? I could not find a match in the MS. PCA only explain the total of ~30+20% of the results. Explain and correct this.
I recommend the change of title.
Author Response
Response to the Reviewer 1 is in the attached file.

Reviewer 2 Report
A great effort to be made, the English language should be reviewed, and the research should be published after review.

Author Response
The response to the rewiewer 2 is in the attached file.

Reviewer 3 Report
The authors compiled a data pool of primary production and environmental variables obtained in Siberia seas in order to develop an empirical algorithm for calculating the optical assimilation number (Pbopt). According their analyses, Pbopt was strongly correlated with surface irradiance; therefore, the new algorithm was based on this environmental variable. The performance of the algorithm was tested by using satellite information. The authors gathered a valuable dataset and the results are promising and useful. However, I think that the manuscript must be improved in several aspects:
(1) The main flaw of the manuscript is that some relevant information on the methodology is missed which does not permit assessing the robustness of the conclusions (see my specific comments).
(2) Conceptually, I think that the physiological reason behind the relationship between Pbopt and irradiance is unclear or appropriately discussed in the manuscript. I think that Pbopt is a parameter that depends on the photoacclimation story of the phytoplankton. Irradiance changes during the daily cycle or more rapidly depending on the weather conditions as well as with depth and water column transparency. The relationship between Pbopt and irradiance obtained is probably reflecting changes in phytoplankton acclimating to the seasonal cycle as I suppose that changes in irradiance run parallel with the seasonal cycle. Other factors would be important in explaining the variability of Pbopt as like as temperature and DIN (which is evident from the results of the PCA). However, it is tentatively discussed in the manuscript. Consequently, I recommend improving the interpretation of the statistical analysis results and reinforcing the discussion accordingly.
(3) The authors presented estimations of the performance of their algorithm but they did not discuss if these deviations between measured and calculated values of Pbopt are suitable taking into account that the objective of the parameterizations of Pbopt is obtaining values of primary production. In that sense, the manuscript would be strongly reinforced if values of IIP based on Pbopt measured and estimated are presented.
Specific comments
Introduction
(4) First paragraph. I am not sure that the term “element” is suitable in this context
Second paragraph. I wonder why the assimilation number is important only to estimate “the annual value of PP”. I suppose that these “mathematical equations” are models for fitting the photosynthesis vs. Irradiance curves (please, be more precise).
(5) Fifth paragraph. It sould read “euphotic layer” better than “photosynthetic layer”.
(6) Sixth paragraph. PAR is “photosynthetically active radiation”. I guess that Pbopt is mainly related to the light story of the phytoplankton (acclimated to low or high irradiance) more than to the specific PAR occurring at a given time.
Material and methods
Material and Methods
Section 2.2
(7) The methods used to estimation of PP and calculation of Pbopt from the data of PP obtained at different depths should be described. I wonder at which depths were the incubations performed, how long they lasted, how irradiances were measured at that depths, how Pbopt was determined from these profiles.
(8) It is also confused which values of irradiance were used for the analyses and comparisons with Pbopt (irradiance occurring during the incubations, daily mean irradiance). This point is essential to assess the results.
(9) The manuscript is also confuses with the nutrient data. I wonder which specific concentration values were used in performing the comparisons with the data Pbopt (means of concentrations in the water column, concentration at the exact depth where Pbopt was measured). Similar comments might be done for the other variables.
Section 2.3.
(10) I think that the Figure 2 is not necessary.
(11) The expression “differences between independent data samples” requires to be rewritten.
Section 2.5
(12) The information about the satellite data is much reduced. The products used should be listed, the time period of the data, its spatial resolution, the downloaded date and version of the processing are not indicated; the procedure used to assess match-up sites is not described; it is not indicated which was the delay between sampling and satellite overpass time was considered suitable. Additionally, many of the stations were located in close proximity to the coast, where the images provided by the satellite are not usable. Furthermore, the presence of high concentration of optically active substances would do inappropriate the standard algorithms used for retrying properties of the water column. I wonder how the authors managed these points.
Results
(13) I suggest that the authors present a Table summarizing the environmental variables just to show which their variability range is. From the Figure 4, with the x-axis scale in log, it is difficult for the reader to imagine how the system studied compares with other places.
(14) In the PCA analysis, most of variability is explained by the salinity gradient; consequently, the relationships between Pbopt and the environmental variables might be partially masked. I suggest performing two PCAs analyses with the pools of So>25 and So<25. Additionally, the contribution of DIN to PC2 appears to be important; in fact, it was negatively correlated to Pbopt. It should be mentioned and discussed.
(15) Fig.6. I guess that the same data of measured Pbopt were used for the comparisons with Eo measured and satellite Eo. However, the x-axes of panel a and b have different scales and, it appears that the data that are being compared are not the same (values higher than 4 mgC(mgChla)-1 h-1 are not plotted in panel a). Similar comment for panels b and c in Figure 7.
(16) Section 3.4. The authors also should show the spatial distribution of their Pbopt values to assess the performance of the algorithm.
Discussion
Section 4.1.
(17) I think that the sentence commencing with “This finding is consistent ...” is contradictory. Regarding to the “effective absorption capacity” I suppose that the authors are mentioning the specific chlorophyll a light absorption coefficient.
Section 4.2.1
(18) Please, see my previous general comment #2.
Section 4.2.5
(19) Please, see my previous comment #12
Author Response
The response to the reviewer 3 is in the attached file.

Reviewer 4 Report
This study investigates the spatiotemporal variability of the “optimal assimilation number” of the total and size-fractionated phytoplankton and its relationship with environmental factors in the Siberian seas based on satellite observations. Furthermore, an empirical region-specific algorithm was developed, and used to give better estimates of the integrated primary production in the Siberian seas. The results are interesting and basically solid, and the manuscript is well organized. I think this manuscript would meet the standards of the Journal of Marine Science and Engineering, if the following issues are addressed.
1. Introduction
- Line 2, “… of the net autotrophic production of the Earth [e.g., 1]”, delete “e.g. ”.
- Line 14, “… described by mathematical equations [e.g., 9, 10].”, delete “e.g. ”.
2. Materials and methods
- Section 2.1, Line 9, “The field data were obtained in summer …”, to “The field data were obtained in boreal summer …”
- Figure 2, the font size of the tick-labels of the x- and y- axes, and the legends are too small, and should be enlarged.
3. Results
- Section 3.2, Line 4-8, the confidence level (p-values) should be given when mentioning the correlation coefficients of R=0.6, R=-0.21 and -0.2, and R=0.23.
4. Discussion
- Section 4.1, in terms of spatiotemporal variations, I would like to suggest the authors to show time series here. Without timeseries analysis, it is hardly to reveal temporal variations.
- Section 4.2, in terms of the influences of environmental factors on Pbopt, the PCA method could also give useful information. I think may be the authors could add these potential influence factors (i.e., PAR, T0, E0, different nutrients, S0, etc.) when they carried out the PCA method in Figure 5a. For example, if two factors are orthogonal to each other in the PCA map, suggesting that they are not correlated; if they are in the same/opposite direction, suggesting that they are positive/negative correlated (e.g., Trombetta et al. 2019; Wang et al. 2022).
Trombetta, T., Vidussl, F., Mas, S., Parin, D., Simier, M., and Mostajir, B. (2019). Water temperature drives phytoplankton blooms in coastal waters. PloS One 14 (4), e0214933. doi: 10.1371/journal.pone.0214933
Wang D, Fang G, Jiang S, Xu Q, Wang G, Wei Z, Wang Y and Xu T
(2022) Satellite-detected phytoplankton blooms in the Japan/East Sea during the past two decades: Magnitude and timing. Front. Mar. Sci. 9:1065066. doi: 10.3389/fmars.2022.1065066
Author Response
The response to the Reviewer 4 see in the attached file.

Round 2
Reviewer 3 Report
I thank the authors for replying my comments and modifying the manuscript accordingly. I have not further comments.